# IO-Adam: Rethinking Memory-Efficient Adaptive Optimizers from Gradient Computation

Yiting Chen[1]  Zongwei Huo[1]  Junchi Yan[1]

## Abstract

Adaptive Moment Estimation (Adam) is one of the most popular and often the default stochastic optimizers for deep neural network training. Using first- and second-moment estimation, Adam provides adaptive learning rates for each parameter, significantly outperforming Stochastic Gradient Descent (SGD). However, as deep neural networks become larger, estimating the first and second moments consumes substantial memory. It motivates various methods to reduce memory usage for adaptive optimizers. In this paper, we propose to rethink the first and second moment estimation from a gradient computation perspective. The gradient of the weight matrix is the multiplication of the input and the gradient of the output. Instead of finding low-rank approximations of the first and second moments, as in previous work, we propose tracking the input and output gradients to efficiently estimate moments. We provide analyses of the similarities and differences between our proposed method, the widely used Adam optimizer, and previous memory-efficient optimizers designed to reduce memory usage. We conduct experiments to verify the effectiveness of our method, which reduces memory usage by up to 30% while preserving similar performance or even improving the performance of Adam.[1]

## 1. Introduction

Deep learning has achieved remarkable success across a wide range of domains, driven in large part by the design of effective optimizers for model training. From Stochastic Gradient Descent (SGD) (Robbins & Monro, 1951) to adaptive learning rate methods such as AdaGrad (Duchi et al., 2011) and RMSprop (Hinton et al., 2012), researchers have sought to accelerate convergence and improve performance. Specifically, Adaptive Moment Estimation (Adam) (Kingma & Ba, 2015) estimates the first and second moments of the gradient and computes an adaptive learning rate for each parameter, combining the advantages of AdaGrad and RMSprop. Due to its fast convergence and minimal need for hyperparameter tuning, Adam has become one of the most popular optimizers. Despite its satisfactory performance, maintaining estimates of the first and second moments of the gradient requires twice as much parameter memory. As neural networks become larger, the substantial memory usage of Adam motivates the development of memory-efficient adaptive optimizers. A typical way to reduce memory usage is to find a low-rank approximation for the estimation of the first or second moment of the gradient. Adafactor (Shazeer & Stern, 2018) proposes to remove the first moment estimation and estimate the second moment with only the per-row and per-column sums of the square gradients. There are various approaches (Gooneratne et al., 2020; Huang et al., 2023; Modoranu et al., 2023; Cosson et al., 2023) that analyze and utilize low-rank approximations of weight gradients. More recently, Galore (Zhao et al., 2024) proposes to project gradients to a low-dimensional subspace based on singular value decomposition (SVD) and apply adaptive moment estimation on the projected gradients.

From a gradient-computation perspective, this paper proposes investigating and rethinking previous memory-efficient approaches that use low-rank gradient approximation. Without loss of generality, we consider the fundamental component of most neural networks, a linear transformation with a weight matrix. Through back propagation, the gradient of the weight matrix is calculated by multiplying the input by the gradient of the output. It reveals a key structural property: the weight gradient can be naturally decomposed into the product of the activation from the previous layer (input) and the error signal from the next layer (output gradient), where the number of samples determines the rank of the gradient. This gradient decomposition of each sample's contribution to the weight gradient provides a principled, fine-grained view for rethinking how optimizer states can be represented and updated more efficiently. Un-

[1]Shanghai Jiao Tong University, Shanghai, China. Correspondence to: Junchi Yan <yanjunchi@sjtu.edu.cn>.

*Proceedings of the 43rd International Conference on Machine Learning*, Seoul, South Korea. PMLR 306, 2026. Copyright 2026 by the author(s).

[1]The code is available in https://github.com/Ytchen981/IO_Adam

like previous approaches that decompose the gradients, we propose tracking the input and output gradients before computing the weight gradient and using these two components to estimate the first and second moments of the weight gradient. By separately storing the input and output gradients instead of the moment estimation in full-matrix form, we reduce the memory usage while preserving an accurate moment estimation.

For the connection and difference between our method and the Adam optimizer, we show that the second moment estimation from our method is the upper bound of that in Adam, which also provides the same regret bound on the convergence rate as that for Adam. We further analyze the difference brought by the larger second-moment estimation from our method. A larger second-moment estimation reduces the element-wise learning rate, which means it can handle a larger learning rate with this discrepancy. We introduce two complementary adjustments to narrow the gap further in a more principled manner. First, we incorporate a lightweight buffering mechanism that accumulates input and output gradients across multiple batches, producing more accurate second-moment estimation and more closely approximating the moment estimation in Adam. Second, inspired by Hölder's inequality, we adjust the exponents applied to the input and output gradients, thereby yielding a tighter upper bound on the second-moment estimate. To verify its effectiveness, we conduct experiments on various language and vision datasets, where our method achieves performance comparable to that of the Adam optimizer while using nearly half as much memory. **The contributions are:**

- We introduce a novel memory-efficient adaptive optimizer. Unlike prior gradient-factorization approaches, we propose tracking the input and output gradients for first- and second-moment estimation, reflecting the intrinsic structure of the gradient.
- We provide analyses of the difference and connection between the proposed optimizer and the widely used Adam optimizer. We show that our estimated second-order moment is an upper bound of that in the Adam optimizer and provide a similar regret bound on the convergence rate.
- We conduct extensive experiments to verify the effectiveness of our proposed memory-efficient optimizer. Our optimizer achieves substantial memory savings, reducing memory usage by up to $30\%$, while preserving or improving upon Adam's optimization performance.

## 2. Related Works

**Adaptive Moment Estimation (Adam).** Adam (Kingma & Ba, 2015) is a widely adopted adaptive learning rate optimizer, which combines the advantages from Ada-

*Table 1.* Comparison between optimizers. We provide a description of the complexity of second-moment estimation.

| Method | Description | Complexity |
|---|---|---|
| Adam (Kingma & Ba, 2015) | Estimate the first moment and second moment of weight gradient by exponential moving average of the weight gradient and the squared weight gradient. | $mn$ |
| Adafactor (Shazeer & Stern, 2018) | Remove first moment estimate, use the per-column sum and per-row sum of the squared weight gradient for the second moment estimate. | $m + n$ |
| Galore (Zhao et al., 2024) | Project gradients into subspaces and estimate the first and second moments in the subspace. | $mr + nr$, $r$ is the rank. |
| IO-Adam (Ours) | We track the input and output gradient before gradient computation and estimate moments with the input and output gradient. | $mb + nb$, $b$ is the buffer size. |

Grad (Duchi et al., 2011) and RMSprop (Hinton et al., 2012). Maintaining exponential moving averages of both the weight gradient itself and the squared weight gradient, Adam dynamically adjusts individual learning rates for each parameter. We provide the Adam algorithm in Alg. 1. Thanks to its fast convergence and minimal requirement for hyperparameter tuning, the Adam optimizer has become the default optimizer in many scenarios. Following the wide adoption of Adam, many works have sought to deepen the theoretical understanding of its behavior (Xie et al., 2022; Kunstner et al., 2023; Dereich & Jentzen, 2024) and to address its limitations through refinement on the algorithm (Reddi et al., 2019; Luo et al., 2019; Xie et al., 2022). Specifically, the first- and second-moment estimation of the Adam optimizer requires twice as much memory as the parameters, motivating approaches to reduce memory usage.

**Memory Efficient Approaches for Adaptive Optimizers.** Various approaches have been introduced to reduce the memory usage. A pioneering approach is Adafactor (Shazeer & Stern, 2018), which introduces a factorized second moment estimation using the per-row and per-column sums of the square gradients. (Dettmers et al., 2021; Li et al., 2023a;b) propose to quantize the optimizer states to reduce the memory usage, while (Lv et al., 2023) proposes to fuse the back propagation and the optimizer update such that the gradient is released right after its computation. Notably, a branch of work focuses on finding a low-rank approximation of the gradient, where the low-rank property of the gradient has been studied and used to reduce the training cost (Shazeer & Stern, 2018; Zhao et al., 2024). Most recently, Galore (Zhao et al., 2024) proposes projecting the gradient to low-rank subspaces and estimating moments for the projected gradients. AdamSNSM (Nguyen & Nguyen, 2024) also proposes the subset norm and subspace momentum, estimating the moment in the subspace. Unlike previous approaches, this paper proposes rethinking gradient decomposition from a gradient-computation perspective, where one of the most natural decompositions of the weight gradient is into the input and output gradients. From this perspective, we pro-

**Algorithm 1** Adam (Kingma & Ba, 2015)

**Input:** Loss function $\mathcal{L}$, trainable parameters $\theta = \{W_1, W_2, \cdots, W_n\}$, step size $\alpha$, hyperparameters $\beta_1$, $\beta_2$, and $\epsilon$
1: Initialize $M_{W_i}^0 = 0$, $V_{W_i}^0 = 0$;
2: **for** $t = 1$ to $T$ **do**
3:     **for** $W_i \in \theta$ **do**
4:         $G_{W_i}^t \leftarrow \nabla_{W_i}\mathcal{L}$;
5:         $M_{W_i}^t \leftarrow \beta_1 M_{W_i}^{t-1} + (1 - \beta_1) \cdot G_{W_i}^t$;
6:         $V_{W_i}^t \leftarrow \beta_2 V_{W_i}^{t-1} + (1 - \beta_2) \cdot (G_{W_i}^t)^2$;
7:         $\hat{M}_{W_i}^t \leftarrow M_{W_i}^t / (1 - \beta_1^t)$;
8:         $\hat{V}_{W_i}^t \leftarrow V_{W_i}^t / (1 - \beta_2^t)$;
9:         $W_i \leftarrow W_i - \alpha \hat{M}_{W_i}^t / (\sqrt{\hat{V}_{W_i}^t} + \epsilon)$;
10:     **end for**
11: **end for**

**Algorithm 2** IO-Adam (ours)

**Input:** Loss function $\mathcal{L}$, trainable parameters $\{W_1, W_2, \cdots, W_n\}$ and corresponding input and output at every step $\{X_W^t, Y_W^t\}$, step size $\alpha$, buffer size $b$, hyperparameters $\beta_1$, $\beta_2$, and $\epsilon$.
1: Initialize $M_{W_i}^0 = 0$, $C_{W_i}^0 = 0$, $R_{W_i}^0 = 0$;
2: **for** $t = 1$ to $T$ **do**
3:     **for** $W_i \in \theta$ **do**
4:         $M_{W_i}^t \leftarrow \beta_1 M_{W_i}^{t-1} + (1 - \beta_1) \cdot (\nabla_{Y_{W_i}^t}\mathcal{L}) X_{W_i}^{t\top}$;
                  ▷ Fused moment update.
5:         $C_{W_i}^t \leftarrow \beta_2 C_{W_i}^{t-1} + (1 - \beta_2) X_{W_i}^{t2} (1_{bs} e_{(t \bmod b)}^\top)$;
6:         $R_{W_i}^t \leftarrow \beta_2 R_{W_i}^{t-1} + (1 - \beta_2)(\nabla_{Y_{W_i}^t}\mathcal{L})^2 (1_{bs} e_{(t \bmod b)}^\top)$;
        ▷ Separately track input and output gradient.
7:         $V_{W_i}^t = C_{W_i}^t R_{W_i}^{t\top}$;
8:         $\hat{M}_{W_i}^t \leftarrow M_{W_i}^t / (1 - \beta_1^t)$;
9:         $\hat{V}_{W_i}^t \leftarrow V_{W_i}^t / (1 - \beta_2^t)^2$;
                 ▷ Modified bias correction.
10:     $W_i \leftarrow W_i - \alpha \hat{M}_{W_i}^t / (\sqrt{\hat{V}_{W_i}^t} + \epsilon)$;
11:     **end for**
12: **end for**

pose a novel memory-efficient adaptive optimizer, namely adaptive moment estimation by input and output gradient (IO-Adam). We provide a comparison between our method and previous works in Table 1.

# 3. Reducing Memory by Tracking Input and Output Gradient

## 3.1. Preliminaries

Without loss of generality, we consider the fundamental component of neural networks: a linear transformation[2] $\mathbf{y} = W\mathbf{x}$, where $\mathbf{y} \in \mathbb{R}^n, \mathbf{x} \in \mathbb{R}^m$ is the input and output and $W \in \mathbb{R}^{n \times m}$ is the weight matrix. For a loss $\mathcal{L}$, the gradient of the weight $W$ is calculated through back propagation:

$$\nabla_W \mathcal{L} = \nabla_{\mathbf{y}} \mathcal{L} \otimes \mathbf{x} = (\nabla_{\mathbf{y}} \mathcal{L}) \mathbf{x}^\top. \tag{1}$$

Since the input $\mathbf{x}$ and output gradient $\nabla_{\mathbf{y}} \mathcal{L}$ are vectors, the gradient is of rank 1. Similarly, when the inputs are in batches with $Y = WX$ where $Y = (\mathbf{y}_1, \mathbf{y}_2, \cdots, \mathbf{y}_{bs}) \in \mathbb{R}^{n \times bs}, X = (\mathbf{x}_1, \mathbf{x}_2, \cdots \mathbf{x}_{bs}) \in \mathbb{R}^{m \times bs}$ and $bs \in \mathbb{R}$ is the batch size, we have:

$$\nabla_W \mathcal{L} = (\nabla_Y \mathcal{L}) X^\top = \sum_i^{bs} \nabla_{\mathbf{y}_i} \mathcal{L} \otimes \mathbf{x}_i, \tag{2}$$

where the rank of the weight gradient is equal to the batch size $bs$. From a gradient-computation perspective, a natural decomposition of the weight gradient is into the input and output gradients. Motivated by this perspective, we propose separately tracking input and output gradients to enable a memory-efficient adaptive optimizer.

---

[2]Many other operations, such as convolution, can also be converted to a linear layer by rearranging the weights and inputs.

## 3.2. The Second Moment Estimation

The second moment estimate adjusts the learning rate for each parameter adaptively. In Adam, the estimation is achieved with the exponential moving average of the squared weight gradient:

$$V_W^t = \beta_2 V_W^{t-1} + (1 - \beta_2) \cdot (\nabla_W \mathcal{L})^2, \tag{3}$$

where $V_W^t$ is the second moment estimation at the $t$-th step and $\beta_2$ is the coefficient. The essence of the second moment estimate is that the learning rate would be adaptively changed for each parameter, where the parameters with larger gradient norm would have a smaller learning rate. From the gradient calculation perspective, as in Eq. 1, parameters that generally receive larger inputs or error signals (output gradient) would have a smaller learning rate. Therefore, we propose to track the squared input and the squared output gradient for the estimation of the second moment.

$$\begin{aligned} \mathbf{c}_W^t &= \beta_2 \mathbf{c}_W^{t-1} + (1 - \beta_2) \cdot X^2 1_{bs}, \\ \mathbf{r}_W^t &= \beta_2 \mathbf{r}_W^{t-1} + (1 - \beta_2) \cdot (\nabla_Y \mathcal{L})^2 1_{bs}, \\ V_W^t &= \mathbf{r}_W^t \mathbf{c}_W^{t\top}. \end{aligned} \tag{4}$$

$1_{bs} \in \mathbb{R}^{bs}$ represents an all-one vector with $bs$ elements. $\mathbf{c}_W^t \in \mathbb{R}^m$ and $\mathbf{r}_W^t \in \mathbb{R}^n$ are the exponential moving averages of the sum of the squared input and the squared output gradient over a batch. We emphasize the connection and difference between our method and the Adafactor (Shazeer & Stern, 2018). In Adafactor, the second moment estimate comprises the squared gradient's per-row sum and per-column sum. The per-row sum contains the information from the squared output gradient, while the per-column sum contains the information from the squared input.

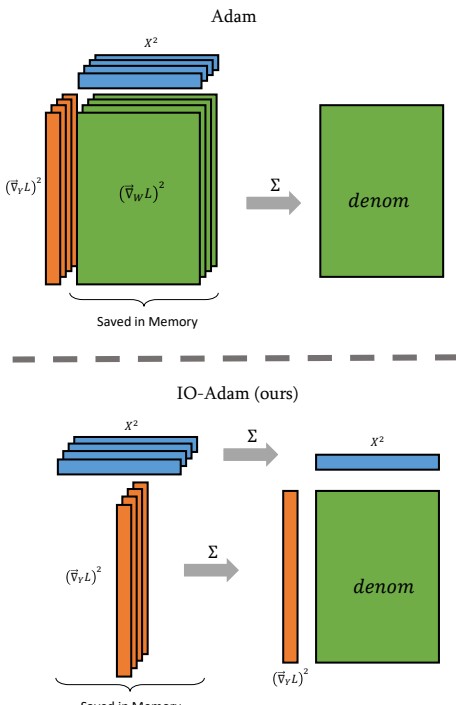

Figure 1. An illustration of the difference between our method and Adam on the second moment estimation.

Specifically, when batch size is 1, the per-row sum and per-column sum in Adafactor are equivalent to $\|\mathbf{x}\|^2 \cdot \nabla_{\mathbf{y}}\mathcal{L}^2$ and $\|\nabla_{\mathbf{y}}\mathcal{L}\|^2 \cdot \mathbf{x}^2$. Instead of operating on the weight gradient, we separately track the input and output gradients, avoiding the variance brought by mixing the squared values from the input and output gradients.

Generally, the second-moment estimate in our method is larger than that in Adam, indicating that our method requires a higher learning rate to achieve similar performance. The reasons are twofold. Firstly, the second-moment estimate in Adam tracks the square of the sum across a batch of samples, whereas our method tracks the sum of squared values. By Cauchy-Buniakowsky-Schwarz Inequality, our second moment estimate is the upper bound of that in Adam. We provide a more detailed analysis in Sec. 3.4. Secondly, the $C_W^t$ and $R_W^t$ in Eq. 4 accumulate values across different batches, multiplying them to introduce the product of input and output gradient from different batches. To better approximate the second moment estimate in Adam, we propose to maintain two matrices instead of vectors with $C_W^t \in \mathbb{R}^{m \times b}$ and $R_W^t \in \mathbb{R}^{n \times b}$, where the columns in the matrices are updated alternately. This means that each column is updated for every $b$ batch, reducing the product of input and output gradients from different batches. Here, we rewrite Eq. 4 as:

$$
\begin{aligned}
C_W^t &= \beta_2 C_W^{t-1} + (1-\beta_2) \cdot X^2 (1_{bs} e_{(t \bmod b)}^\top), \\
R_W^t &= \beta_2 R_W^{t-1} + (1-\beta_2) \cdot (\nabla_Y \mathcal{L})^2 (1_{bs} e_{(t \bmod b)}^\top), \quad (5) \\
V_W^t &= C_W^t R_W^{t\top}.
\end{aligned}
$$

Let $e_{(t \bmod b)} \in \mathbb{R}^b$ denote a standard basis vector with its $(t \bmod b)$-th component equal to 1 and all other components are 0, which controls which column of $C$ and $R$ to be updated. Since $C$ and $R$ are separately updated, the bias correction is also modified by separately applying bias correction on the second moment estimation on the input and output gradient. We present our method in Algorithm 2.

### 3.3. A Possible Buffer for the First Moment Estimation

**Fused Momentum Update.** The first moment estimate, or the momentum, is introduced to accelerate the convergence of stochastic optimizers. In Adam, the first moment estimation is stored and updated as an exponential moving average of the gradient.

$$
M_W^t = \beta_1 M_W^{t-1} + (1-\beta_1) \cdot \nabla_W \mathcal{L}, \quad (6)
$$

where $M_W^t$ is the momentum at the $t$-th step and $\beta_1$ is the coefficient. Since we propose to track the input and the output gradient before weight gradient calculation, the momentum update could be fused with the gradient calculation, similar to the fused backpropagation as in (Lv et al., 2023). Instead of conducting backpropagation and then updating the momentum, the weight gradient does not need to be stored in memory, where the momentum could be directly updated by

$$
M_W^t = \beta_1 M_W^{t-1} + (1-\beta_1) \cdot (\nabla_Y \mathcal{L}) X^\top. \quad (7)
$$

**Buffering Mechanism.** For scenarios where the batch size is rather small, we could keep a buffer to record the input and the output gradient, such that

$$
\begin{aligned}
B_{\mathbf{x}}^t &= \left((1-\beta_1)\mathbf{x}^t, \beta_1(1-\beta_1)\mathbf{x}^{t-1}, \cdots, \beta_1^b(1-\beta_1)\mathbf{x}^{t-b}\right), \\
B_{\nabla_y \mathcal{L}}^t &= \left((1-\beta_1)\nabla_{y^t}\mathcal{L}, \beta_1(1-\beta_1)\nabla_{y^{t-1}}\mathcal{L}, \cdots, \beta_1^b(1-\beta_1)\nabla_{\mathbf{y}^{t-b}}\mathcal{L}\right).
\end{aligned} \quad (8)
$$

$B_{\mathbf{x}}^t$ and $B_{\nabla_y \mathcal{L}}^t$ represent the buffer and $\mathbf{x}^t$ and $\nabla_{y^t}\mathcal{L}$ is the input and output gradient at the $t$-th step. By tracking the most recent $b$ batches, the moment estimation is provided by

$$
M_W^t = B_{\nabla_y \mathcal{L}}^t B_{\mathbf{x}}^{t\top}. \quad (9)
$$

As long as $b \cdot bs \cdot (m+n) < m \cdot n$, keeping the buffer would reduce the memory usage. Note that, when $b$ is set to 1, it is equivalent to an optimizer without momentum, such as the Adafactor (Shazeer & Stern, 2018). Given the large scale of modern neural networks, separately storing the input and output gradients could still generally reduce the memory usage compared to storing the whole weight gradient.

**Remark:** In experiments on LLMs, we do not adopt the buffer for first moment estimation, since each data sample contains numerous tokens that may require a large buffer. As demonstrated in prior works (Shazeer & Stern, 2018),

the first moment estimation may not have a significant effect on the performance of an optimizer. Therefore, we provide only an analysis of the possible buffer for first-moment estimation in scenarios with less inputs.

### 3.4. The Connection and Difference between Our Method and Adam

The primary difference between our method and Adam lies in the estimation of the second moment, where Adam utilizes the squared gradient, whereas our method employs the squared input and output gradients. In this section, we analyze the connection between our method and Adam and further investigate the theoretical convergence properties.

#### 3.4.1. THE UPPER BOUND OF THE SECOND MOMENT ESTIMATE

In this section, we show that our second moment estimate is the upper bound of the second moment estimate in Adam. Let $\frac{\partial \mathcal{L}}{\partial \mathbf{y}_{ij}}$ denotes the $j$-th element in $\nabla_{\mathbf{y}_i} \mathcal{L}$ and $\mathbf{x}_{ij}$ denotes the $j$-th element in $\mathbf{x}$. For the weight gradient, with Eq. 2 we have:

$$\nabla_W \mathcal{L} = \begin{pmatrix} \sum_i^{bs} \frac{\partial \mathcal{L}}{\partial \mathbf{y}_{i1}} \cdot \mathbf{x}_{i1} & \sum_i^{bs} \frac{\partial \mathcal{L}}{\partial \mathbf{y}_{i1}} \cdot \mathbf{x}_{i2} & \cdots & \frac{\partial \mathcal{L}}{\partial \mathbf{y}_{i1}} \cdot \mathbf{x}_{im} \\ \sum_i^{bs} \frac{\partial \mathcal{L}}{\partial \mathbf{y}_{i2}} \cdot \mathbf{x}_{i1} & \sum_i^{bs} \frac{\partial \mathcal{L}}{\partial \mathbf{y}_{i2}} \cdot \mathbf{x}_{i2} & \cdots & \frac{\partial \mathcal{L}}{\partial \mathbf{y}_{i2}} \cdot \mathbf{x}_{im} \\ \vdots & \vdots & \ddots & \vdots \\ \sum_i^{bs} \frac{\partial \mathcal{L}}{\partial \mathbf{y}_{in}} \cdot \mathbf{x}_{i1} & \sum_i^{bs} \frac{\partial \mathcal{L}}{\partial \mathbf{y}_{in}} \cdot \mathbf{x}_{i2} & \cdots & \frac{\partial \mathcal{L}}{\partial \mathbf{y}_{in}} \cdot \mathbf{x}_{im} \end{pmatrix}.$$
(10)

For the element $(\nabla_W \mathcal{L})_{[j,k]}$ at the $j$-th column and the $k$-th row of the matrix, we have:

$$(\nabla_W \mathcal{L})_{[j,k]} = \sum_i^{bs} \frac{\partial \mathcal{L}}{\partial \mathbf{y}_{ij}} \cdot \mathbf{x}_{ik}.$$
(11)

Similarly, for the squared weight gradient, we have,

$$(\nabla_W \mathcal{L})_{[j,k]}^2 = \left( \sum_i^{bs} \frac{\partial \mathcal{L}}{\partial \mathbf{y}_{ij}} \cdot \mathbf{x}_{ik} \right)^2.$$
(12)

In our method, we take the sum of the squared input and output gradients, where

$$X^2 1_{bs} = \left( \sum_i^{bs} \mathbf{x}_{i1}^2, \quad \sum_i^{bs} \mathbf{x}_{i2}^2, \quad \cdots, \sum_i^{bs} \mathbf{x}_{im}^2 \right)^\top,$$

$$(\nabla_Y \mathcal{L})^2 1_{bs} = \left( \sum_i^{bs} (\frac{\partial \mathcal{L}}{\partial y_{i1}})^2, \quad \sum_i^{bs} (\frac{\partial \mathcal{L}}{\partial y_{i2}})^2, \quad \cdots, \sum_i^{bs} (\frac{\partial \mathcal{L}}{\partial y_{in}})^2 \right)^\top.$$
(13)

Our second moment estimate is based on the outer product between the two vectors in Eq. 13. The element at the $j$-th column and the $k$-th row of the outer product is given by

$$\left( (\nabla_Y \mathcal{L})^2 1_{bs} \otimes X^2 1_{bs} \right)_{[j,k]} = \left( \sum_i^{bs} (\frac{\partial \mathcal{L}}{\partial \mathbf{y}_{ij}})^2 \right) \cdot \left( \sum_i^{bs} \mathbf{x}_{ik}^2 \right).$$
(14)

Comparing Eq. 12 to Eq. 14, by the Cauchy-Buniakowsky-Schwarz Inequality, we have

$$\forall j, k : (\nabla_W \mathcal{L})_{[j,k]}^2 \leq \left( (\nabla_Y \mathcal{L})^2 1_{bs} \otimes X^2 1_{bs} \right)_{[j,k]}.$$
(15)

Eq. 15 indicates that the outer product of squared input and squared output gradient is the upper bound of the squared weight gradient. Therefore, the second moment estimate in our method is larger than the second moment estimate in the Adam optimizer.

#### 3.4.2. CONVERGENCE ANALYSIS

Following Adam (Kingma & Ba, 2015) and the adjustment pointed out in AMSgrad (Reddi et al., 2019), we provide a similar theoretical analysis on the convergence. Based on the online learning framework (Zinkevich, 2003), the optimizer aims to predict the parameter $\theta_t$ with an arbitrary, unknown sequence of the convex cost functions $f_1(\theta), f_2(\theta), \cdots, f_T(\theta)$ and evaluate the parameter on a previously unknown cost $f_t$. The algorithm is evaluated with the regret: the sum of all previous differences between the online prediction $f_t(\theta_t)$ and the best fixed point parameter $f_t(\theta^*)$ for all previous steps. The regret is defined as:

$$R(T) = \sum_{t=1}^{T} [f_t(\theta_t) - f_t(\theta^*)],$$
(16)

where $\theta^* = \arg\min_\theta \sum_{t=1}^{T} f_t(\theta)$. The theoretical analysis in (Reddi et al., 2019) shows that Adam has $O(\sqrt{T})$ regret bound when the cost function $f_t$ has bounded gradients, such that $\lim_{T \to \infty} \frac{R(T)}{T} = 0$ when the second moment estimation is monotonically increasing (the variant of Adam, AMSgrad (Reddi et al., 2019)). We show that our method has a similar regret bound. The proof is in the Appendix A.

**Proposition 3.1.** *[Regret Bound] Assume that the cost function $f_t$ has bounded gradients, $\forall \theta : \|\nabla f_t(\theta)\|_2 \leq G, \|\nabla f_t(\theta)\|_\infty \leq G_\infty$ and distance between $\theta_t$ generated by our method is bounded, $\forall m, n \in \{1, 2, \cdots, T\} : \|\theta_m - \theta_n\|_2 \leq D, \|\theta_m - \theta_n\|_\infty \leq D_\infty$. It achieves the guarantee:*

$$\forall T \geq 1 : \frac{R(T)}{T} = O(\frac{1}{\sqrt{T}}).$$

Proposition 3.1 states that our method also has $O(\sqrt{T})$ regret bound such that $\lim_{T \to \infty} \frac{R(T)}{T} = 0$. The proof of Proposition 3.1 is similar to the analysis provided in the Adam paper (Kingma & Ba, 2015). Generally, our second moment estimate is larger than the second moment estimate in Adam while also being bounded when the gradient is bounded, which provides us with a similar regret bound as Adam.

*Table 2.* Results of fine-tuning RoBERTa-Base (Liu et al., 2019) on GLUE (Wang et al., 2018). We use the same hyperparameters across all the tasks. The peak memory allocation data is collected on CoLA with a batch size of 16.

| Methods | CoLA | STSB | MRPC | RTE | SST2 | MNLI | QNLI | QQP | Mem (MB) |
|---|---|---|---|---|---|---|---|---|---|
| AdamW (Loshchilov & Hutter, 2019) | **62.82** | 90.67 | **92.17** | 74.01 | 92.09 | 86.59 | 91.51 | 91.46 | 2409.89 |
| Adafactor (Shazeer & Stern, 2018) | 58.08 | 90.25 | 91.03 | 74.45 | 93.15 | 87.04 | **92.84** | 89.93 | 1133.41 |
| Galore | 58.54 | 90.66 | 91.81 | 73.65 | 92.66 | 85.80 | 90.83 | 90.74 | 1768.14 |
| IO-Adam (Ours) | 61.59 | **90.68** | 91.30 | **75.09** | **94.04** | **87.12** | 92.40 | **91.68** | 1599.61 |

### 3.4.3. A POSSIBLE MODIFICATION ON OUR METHOD INSPIRED BY HÖLDER'S INEQUALITY

While Sec. 3.4.1 shows that the outer product between the squared input and the squared output gradient is the upper bound of the squared weight gradient, in this section, we discuss a possible approach to tighten the bound. By Hölder's inequality (Hardy & Littlewood), for all $p, q \in [1, \infty], \frac{1}{p} + \frac{1}{q} = 1$ we have,

$$\sum_{i}^{n} |a_i b_i| \leq \left( \sum_{i}^{n} a_i^p \right)^{\frac{1}{p}} \cdot \left( \sum_{i}^{n} b_i^q \right)^{\frac{1}{q}}. \quad (17)$$

The Hölder's inequality is an extension of the Cauchy-Buniakowsky-Schwarz Inequality. When $p = 2, q = 2$, Eq. 17 becomes the Cauchy Inequality. By adjusting the $p$ and $q$, one may tighten the bound. For neural networks, the inputs are generally more sparse, *e.g.*, negative elements become zero after ReLU activation. On the other hand, the output gradients could drastically change, where previous works (Cohen et al., 2021; Damian et al., 2022) show that the loss landscape's sharpness may even increase as the model becomes closer to the minimum. Therefore, to adjust $p$ and $q$ for the input and output gradient, we could use a smaller $p \in [1, 2]$ for the input and leave the $q \in [2, +\infty]$ for the output gradient. By doing so, the second moment estimation in our method, Eq. 4, could be adjusted to:

$$\begin{aligned} \mathbf{c}_W^t &= \beta_2 \mathbf{c}_W^{t-1} + (1 - \beta_2) \cdot |X|^p 1_{bs}, \\ \mathbf{r}_W^t &= \beta_2 \mathbf{r}_W^{t-1} + (1 - \beta_2) \cdot |\nabla_Y \mathcal{L}|^q 1_{bs}, \\ \sqrt{V_W^t} &= (\mathbf{r}_W^t)^{\frac{1}{p}} (\mathbf{c}_W^{t\top})^{\frac{1}{q}}. \end{aligned} \quad (18)$$

This modification introduces a potential hyperparameter $p$, which could be adjusted for better performance when the input or the output gradients are sparse. In Sec. 4.4, we provide experimental results, shedding light on how the $p$ could influence performance. While this is an additional modification, not the main purpose, unless otherwise specified, the $p$ is set to 2 in our experiments.

## 4. Experiments

### 4.1. The Glue Benchmark

We follow the setting in Galore (Zhao et al., 2024) to conduct experiments on GLUE (Wang et al., 2018) with pre-trained RoBERTa-Base (Liu et al., 2019). GLUE is a benchmark for evaluating language models on various tasks, including sentiment analysis, question answering, and textual entailment. With experiments on GLUE, we compare our method with AdamW[3] (Loshchilov & Hutter, 2019), Adafactor (Shazeer & Stern, 2018), and Galore (Zhao et al., 2024). The hyperparameters are set the same for all the tasks. The model is fine-tuned for 30 epochs with the learning rate set at $3e - 5$ and a linear learning rate scheduler. The other settings follow the default settings in PyTorch and Galore for all the optimizers. Our method uses fused first moment estimation and a buffering mechanism for the second moment estimation as introduced in Sec. 3.3 and Sec. 3.2. The buffer size for our method and the rank for Galore are both set to 16. We will provide more details about the hyperparameter setting in the Appendix.

As shown in Table 2, we present the results of fine-tuning RoBERTa-Base on GLUE. Generally, our method achieves similar performance to Adam and outperforms Adafactor and Galore on most tasks. Regarding memory usage, Adafactor achieves the lowest memory usage with no first moment estimation and two vectors for the second moment estimation. Our method generally achieves similar memory reduction as Galore with fused first moment estimation and a buffer for second moment estimation. Note that we track the first moment estimate with a full matrix, which could be combined with Galore, projecting the moment estimate to subspaces for further memory reduction.

### 4.2. Pretraining LLaMA on C4 and MiniPile

To further test our optimizer, we pretrain small LLaMA (Touvron et al., 2023) on the C4 dataset (Raffel et al., 2020) and MiniPile (Kaddour, 2023) following Galore (Zhao et al., 2024) and AdamSNSM (Nguyen & Nguyen, 2024). The Colossal Cleaned Common Crawl (C4) dataset is a large-scale multilingual corpus of text extracted from the web, and MiniPile is a filtered subset of the Pile corpus. Following the setting in Galore (Zhao et al., 2024), we use models of size 60M, 130M and 1B. We compare our method with AdamW (Loshchilov & Hutter, 2019), Adafactor (Shazeer & Stern, 2018), Galore (Zhao et al., 2024), and AdamSNSM (Nguyen & Nguyen, 2024). The rank for Galore and the buffer size in our method are set to the same,

---

[3]Adam with its weight decay disentangled

*Table 3.* The final evaluation perplexity of the LLaMA model (Touvron et al., 2023) of different sizes trained on the C4 dataset, following AdamSNSM (Nguyen & Nguyen, 2024). (* denotes the results referenced from AdamSNSM (Nguyen & Nguyen, 2024).)

| Perplexity ($\downarrow$) | LLaMA-60M | LLaMA-130M | LLaMA-1B |
|---|---|---|---|
| Adam (Kingma & Ba, 2015) | 30.46* | 24.60* | 16.00* |
| AdamW (Loshchilov & Hutter, 2019) | 29.62 | 22.61 | 14.51 |
| Galore (Zhao et al., 2024) | 34.09 | 25.31 | 16.76* |
| AdamSNSM (Nguyen & Nguyen, 2024) | 29.84 | 22.71 | 14.05* |
| IO-Adam | 29.75 | 22.40 | 14.36 |

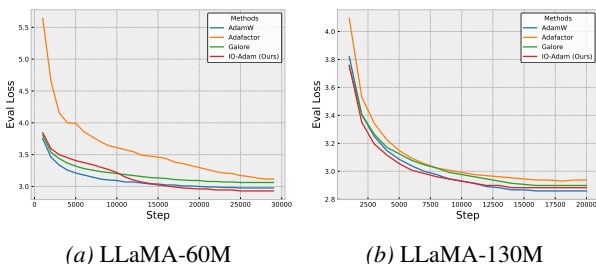

*(a) LLaMA-60M*      *(b) LLaMA-130M*

*Figure 2.* The evaluation loss of LLaMA-60M and 130M on MiniPile. Our method outperforms Adafactor and Galore and achieves a similar performance to AdamW.

128 for LLaMA-60M, 256 for LLaMA-130M, and 1024 for LLaMA-1B. For each optimizer, the learning rate is grid searched in the set $\{0.025, 0.02, 0.01, 0.002, 0.001\}$.

As shown in Figure 2, we report the evaluation loss of the LLaMA-60M and LLaMA-130M through the training progress on MiniPile. Generally, our method achieves a performance similar to or better than that of AdamW. As analyzed in Sec. 3.4, our method yields a larger second moment estimation compared to Adam, which means our method could use a larger learning rate than AdamW. In this pretraining scenario, the optimal learning rate for Adam is 0.002, while the optimal learning rate for our method is 0.025, differing by an order of magnitude. It indicates that in some scenarios, the learning rate for our method should be substantially higher than that for Adam or AdamW.

As shown in Table 3, we report the final evaluation perplexity of LLaMA models trained on the C4 dataset. We follow the setting in AdamSNSM (Nguyen & Nguyen, 2024) to conduct our experiments. The learning rate of our method for the 60M and 130M models is searched in the set $\{0.025, 0.02, 0.01, 0.002, 0.001\}$. The learning rate for the 1B model is set to 0.002 due to the high computational cost. The results in Table 3 demonstrate the effectiveness of our method, where our method achieves a similar performance to AdamW (Loshchilov & Hutter, 2019). Details are in Appendix B.

### 4.3. Vision Transformer on CIFAR10

To evaluate our optimizer on vision tasks, we train vision-transformers (ViT) (Dosovitskiy et al., 2020) on CI-

*Table 4.* Results of training ViT-small on CIFAR10 with different optimizers. We test our method both with and without a buffer for the second moment estimation as mentioned in Sec. 3.2.

| Methods | eval-acc(%) | eval-loss | mem-alloc(MB) |
|---|---|---|---|
| AdamW | 76.06 | 0.76402 | 112.02 |
| Adafactor | 76.32 | 0.75529 | **64.52** |
| IO-Adam(Ours) w/buffer size $b = 16$ | **77.63** | **0.69870** | 68.40 |
| IO-Adam(Ours) w/buffer size $b = 8$ | 76.87 | 0.71221 | 66.84 |
| IO-Adam(Ours) w/buffer size $b = 4$ | 77.01 | 0.70820 | 66.06 |
| IO-Adam(Ours) w/buffer size $b = 2$ | 77.13 | 0.72194 | 65.66 |
| IO-Adam(Ours) w/buffer size $b = 1$ | 75.46 | 0.73762 | 65.47 |

FAR10 (Krizhevsky et al., 2009). We use the code from vit-pytorch to conduct our experiments and compare our method to AdamW (Loshchilov & Hutter, 2019) and Adafactor (Shazeer & Stern, 2018). For each optimizer, we train a ViT-small with default hyperparameters except the learning rate, where the learning rate is set at $5e-4$ for AdamW and $1e-3$ for our method, selected with a grid search. Note that Adafactor is designed not to require a learning rate; in our experiment, we use the default learning rate $1e-2$ in PyTorch. We also conduct experiments to verify the impact of buffer size $b$ for second moment estimation on our method, where we select the buffer size from $\{1, 2, 4, 8, 16\}$. Details about hyperparameter settings are in Appendix B.

The results are shown in Table 4. While the Adafactor achieves the lowest memory usage, our method with a buffer of size 16 achieves the best performance compared to AdamW and Adafactor with a relatively low memory usage. Comparing the performance of our method with different buffer sizes, we show that a buffer for second moment estimation is important for better performance, where a buffer reduces the cross-batch terms when we multiply the moving average of the squared input and the squared output gradient, leading to a more accurate and smaller second moment estimation. Generally, a larger buffer size means that our second-moment estimation contains fewer cross-batch terms corresponding to the product of inputs and output gradients from different batches.

### 4.4. Experiments on WikiText

We conduct experiments on WikiText-2 (Merity et al., 2016) with GPT-2 (Radford et al., 2019). WikiText-2 is a language-modeling benchmark that comprises clean text from English Wikipedia articles. We follow the setting in the official example from the Transformers library (Wolf et al., 2020)

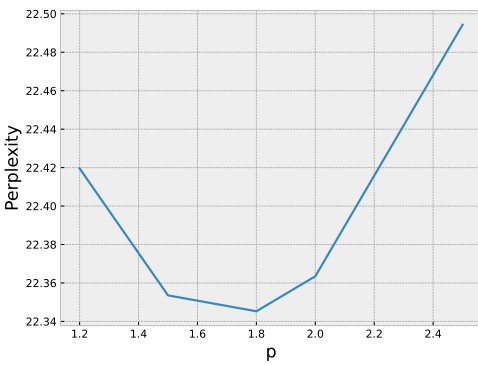

*Figure 3.* The perplexity of models finetuned by IO-Adam with different $p$ as discussed in Sec.3.4.3. According to the result, $p$ around 1.8 achieves the best performance.

*Table 5.* Results of finetuning GPT-2 (Radford et al., 2019) with different optimizers on WikiText-2.

| Methods | Perplexity($\downarrow$) |
|---------|-------------------------|
| AdamW | 22.63 |
| Adafactor | 22.72 |
| IO-Adam (Ours) | **22.36** |

to fine-tune a pretrained GPT-2 on WikiText-2. While GPT-2 employs 1-dimensional convolution, which is basically equivalent to a linear layer, we replace each 1D convolution with a linear layer. Our method, IO-Adam, is compared to AdamW and Adafactor. The learning rate is set at $5e-4$ for IO-Adam and AdamW and $5e-5$ for Adafactor. The buffer size of IO-Adam is 1. The results are shown in Table 5, where our method achieves a lower perplexity compared to Adafactor and AdamW.

**Experiments on the $p$ modification as introduced in Sec. 3.4.3.** We further conduct experiments on the modification of $p$ inspired by Hölder's inequality as introduced in Sec. 3.4.3. By varying $p$, we aim to achieve a more accurate estimate of the second moment. While the output gradient could be more drastically changed, the $p$ should be smaller for the input, leaving a larger $q = \frac{p}{p-1}$ for the output gradient. To test how this modification would affect our method, we fine-tune GPT-2 on WikiText-2 by IO-Adam with different $p$. As shown in Fig. 3, as the $p$ increases from 1.2 to 2.5, the perplexity first decreases and then increases. $p = 1.8$ achieves the lowest perplexity, which means the most proper $p$ is around 1.8 for fine-tuning GPT-2 on WikiText-2 in this scenario. The results corroborate our analysis, indicating that lowering $p$ could further improve performance. While $p$ is a potential hyperparameter, our other experimental results indicate that setting $p$ to 2 generally performs well. With the hyperparameter $p$ introduced, we propose an option to treat the input and output gradients differently for adaptive learning rate adjustment, where the input gradient is generally stable when the learning rate is low, whereas the output gradient can change substantially when the loss landscape

*Table 6.* The average time of each step when training LLaMA-60M on the C4 dataset. The batch size is set to 64, and the gradient is accumulated for eight batches. The experiments are conducted on a L20Z GPU.

| Method | Average Time per step (s) |
|--------|---------------------------|
| AdamW | 0.2817 |
| Adafactor | 0.2892 |
| Galore | 0.2949 |
| IO-Adam (Ours) | 0.3010 |

is sharp. While previous works treat the weight gradient as a whole, our method is unique in its ability to account for differences between the input and output gradients.

### 4.5. Time Overhead of IO-Adam

Generally, memory-efficient optimizers require additional operations to trade time for space. Our method requires tracking the input and output gradients, which introduces a small amount of overhead compared to Adam. The runtime is close to those memory-efficient optimizers such as Galore and Adafactor. In Table 6, we report the average time for each step when training LLaMA-60M on the C4 dataset. While our implementation currently uses hooks in PyTorch to track intermediate hidden states and gradients, it is slightly slower than other methods (approximately 7%). While backward propagation already involves multiplying the input by the output gradient to calculate the weight gradient, it may be possible to improve input and output gradient monitoring using the fundamental gradient calculation mechanism, which could further improve effectiveness.

## 5. Conclusion

In this paper, we propose rethinking previous memory-efficient approaches for adaptive optimizers from the perspective of gradient computation. We show that the input and the output gradient are a natural decomposition of the weight gradient. From this perspective, we propose a memory-efficient adaptive optimizer that separately tracks the input and output gradients for adaptive moment estimation, namely IO-Adam. We analyze the connection and difference between IO-Adam and vanilla Adam, showing that our method has the same regret bound for convergence rate as Adam. We conduct experiments on language modeling and vision tasks, where our method substantially reduces memory usage while maintaining performance comparable to or superior to Adam. The primary limitation of our method stems from structural constraints, as it focuses on the fundamental component, linear layers, in neural networks. Further modifications to our method may be required to apply it to other modules, such as convolution. We hope our work will inspire future works to enhance the effectiveness and efficiency of adaptive optimizers.

## Impact Statement

This work presents a memory-efficient optimization algorithm that reduces the GPU memory footprint during training of large-scale neural networks. The primary contribution is methodological, offering a practical solution to a key technical constraint in modern deep learning. There are many potential societal consequences of our work, none of which we feel need to be highlighted here.

## Acknowledgements

This work was partly supported by Scientific Research Innovation Capability Support Project for Young Faculty (U40) of the Ministry of Education of China, SRICSPYF-ZY2025019

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

# A. Proof for Proposition 3.1

Generally, we follow the analysis in (Kingma & Ba, 2015) for the proof of Proposition 3.1. The main difference between our method and Adam is that we separately track squared input and squared output gradient for the second moment estimation, which will result in a larger second moment estimate. We start with the definition of convex functions.

**Definition A.1.** A function $f : \mathbb{R}^d \to \mathbb{R}$ is convex if for all $x, y \in \mathbb{R}^d$, for all $\lambda \in [0, 1]$,

$$\lambda f(x) + (1 - \lambda)f(y) \geq f(\lambda x + (1 - \lambda)y). \tag{19}$$

**Lemma A.2.** If a function $f : \mathbb{R}^d \to \mathbb{R}$ is convex, for all $x, y \in \mathbb{R}^d$,

$$f(y) \geq f(x) + \nabla f(x)^\top (y - x) \tag{20}$$

Similar to that in (Kingma & Ba, 2015), we use the above lemma to upper bound the regret. Since all the parameters are reshaped to a vector, to align the notation with (Kingma & Ba, 2015), we use $g_t$ to denote $f_t(\theta_t)$ and use $m_t$ to denote the momentum at the $t$-th step updated by $m_t = \beta_1 m_{t-1} + (1 - \beta_1)\nabla f_t(\theta_t)$ and $\hat{m}_t$ is $\frac{m_t}{(1-\beta_1^t)}$. For the second moment estimate, we use $v_t$ to represent our second moment estimate. Specifically $\theta_{t,i}, g_{t,i}, m_{t,i}$, and $v_{t,i}$ are used to denote the $i$-th element of the corresponding vector.

**Lemma A.3.** Let $\gamma \triangleq \frac{\beta_1}{\beta_2}$. For $\beta_1, \beta_2 \in [0, 1]$ that satisfy $\gamma < 1$, and bounded $g_t$ with $\|g_t\| \leq G$, $\|g_t\|_\infty \leq G_\infty$, the following inequality holds,

$$\sum_{t=1}^{T} \frac{\hat{m}_{t,i}^2}{\sqrt{t\hat{v}_{t,i}}} < \frac{2(1 - \beta_2^t)(1 - \beta_1)^2}{(1 - \beta_2)^2(1 - \beta_1^t)^2} G_\infty \sqrt{T}. \tag{21}$$

*Proof.* According to Eq. 15, our second moment estimation is generally larger than that of the Adam,

$$v_{t,i} \geq \sum_{j=1}^{t} (1 - \beta_2)^2 \beta_2^{2(t-j)} g_{j,i}^2. \tag{22}$$

For the first moment estimate, we have

$$m_{t,i}^2 = \left[ \sum_{j=1}^{t} (1 - \beta_1)\beta_1^{t-j} g_{j,i} \right]^2. \tag{23}$$

Applying the Cauchy–Schwarz inequality, we have

$$m_{t,i}^2 \leq \left( \sum_{j=1}^{t} \left[ \frac{(1 - \beta_1)\beta_1^{t-j}}{(1 - \beta_2)\beta_2^{t-j}} \right]^2 \right) \left( \sum_{j=1}^{t} (1 - \beta_2)^2 \beta_2^{2(t-j)} g_{j,i}^2 \right) \tag{24}$$

Combining Eq. 22 and Eq. 24, we further have

$$
\begin{aligned}
\frac{\hat{m}_{t,i}^2}{\sqrt{\hat{v}_{t,i}}} &= \frac{(1 - \beta_2^t)}{(1 - \beta_1^t)^2} \frac{m_{t,i}^2}{\sqrt{v_{t,i}}} \\
&\leq \frac{(1 - \beta_2^t)}{(1 - \beta_1^t)^2} \left( \sum_{j=1}^{t} \left[ \frac{(1 - \beta_1)\beta_1^{t-j}}{(1 - \beta_2)\beta_2^{t-j}} \right]^2 \right) \frac{\sum_{j=1}^{t} (1 - \beta_2)^2 \beta_2^{2(t-j)} g_{j,i}^2}{\sqrt{\sum_{j=1}^{t} (1 - \beta_2)^2 \beta_2^{2(t-j)} g_{j,i}^2}} \\
&\leq \frac{(1 - \beta_2^t)(1 - \beta_1)^2}{(1 - \beta_2)^2(1 - \beta_1^t)^2} \sum_{j=1}^{t} \gamma^{2(t-j)} G_\infty.
\end{aligned} \tag{25}
$$

Since $\gamma < 1$, for all $t \in \{1, 2, \cdots, T\}$ we have

$$\frac{\hat{m}_{t,i}^2}{\sqrt{\hat{v}_{t,i}}} \leq \frac{(1 - \beta_2^t)(1 - \beta_1)^2}{(1 - \beta_2)^2(1 - \beta_1^t)^2} G_\infty. \tag{26}$$

Therefore,

$$
\sum_{t=1}^{T} \frac{\hat{m}_{t,i}^2}{\sqrt{t\hat{v}_{t,i}}} \leq \frac{(1-\beta_2^t)(1-\beta_1)^2}{(1-\beta_2)^2(1-\beta_1^t)^2} G_\infty \sum_{t=1}^{T} \frac{1}{\sqrt{t}}
$$
$$
< \frac{2(1-\beta_2^t)(1-\beta_1)^2}{(1-\beta_2)^2(1-\beta_1^t)^2} G_\infty \sqrt{T}.
$$

(27)

$\square$

**Theorem A.4.** *Assume the convex function $f_t$ has bounded gradients, $\|\nabla f_t(\theta)\|_2 \leq G$, $\|\nabla f_t(\theta)\|_\infty \leq G_\infty$ for all $\theta \in \mathbb{R}^d$ and the $\theta$ generated by IO-Adam is bounded by $\|\theta_n - \theta_m\|_2 \leq D$, $\|\theta_n - \theta_m\|_\infty \leq D_\infty$ for any $n, m \in \{1, 2, \cdots, T\}$, and $\beta_1, \beta_2 \in [0, 1)$ satisfy $\frac{\beta_1}{\beta_2} < 1$. Let $\alpha_t = \frac{\alpha}{\sqrt{t}}$ and $\beta_{1,t} = \beta_1\lambda^{t-1}$, $\lambda \in (0, 1)$. IO-Adam achieves the following guarantee, for all $T > 1$,*

$$
R(T) \leq \frac{dD^2}{\alpha(1-\beta_1)} G_\infty \sqrt{T} + \frac{\alpha d(1-\beta_2^t)(1-\beta_1^2)}{(1-\beta_2)^2(1-\beta_1^t)^2} G_\infty \sqrt{T} + \frac{dD_\infty^2 G_\infty}{\alpha(1-\beta_1)(1-\lambda)^2}.
$$

(28)

*Proof.* Using Lemma A.2, we have

$$
f_t(\theta_t) - f_t(\theta^*) \leq \nabla f_t(\theta_t)^\top (\theta_t - \theta^*).
$$

(29)

According to the update rule in Algorithm 2,

$$
\theta_{t+1} = \theta_t - \alpha_t \hat{m}_t / \sqrt{\hat{v}_t}
$$
$$
= \theta_t - \frac{\alpha_t}{(1-\beta_1^t)\sqrt{\hat{v}_t}} [\beta_{1,t} m_{t-1} + (1-\beta_{1,t})\nabla f_t(\theta_t)].
$$

(30)

With the update rule, we have

$$
(\theta_{t+1} - \theta^*)^2 = (\theta_t - \theta^* - \alpha_t \hat{m}_t / \sqrt{\hat{v}_t})^2
$$
$$
= (\theta_t - \theta^*)^2 - \frac{2\alpha_t \hat{m}_t}{\sqrt{\hat{v}_t}} \odot (\theta_t - \theta^*) + \frac{\alpha_t^2 \hat{m}_t^2}{\hat{v}_t}
$$

(31)

Substitute $\hat{m}_t$ with $\frac{\beta_{1,t} m_{t-1} + (1-\beta_{1,t})\nabla f_t(\theta_t)}{(1-\beta_1^t)}$ in Eq. 31, we have

$$
\nabla f_t(\theta_t) \odot (\theta_t - \theta^*) = \frac{(1-\beta_1^t)\sqrt{\hat{v}_t}}{2\alpha_t(1-\beta_{1,t})}[(\theta_t - \theta^*)^2 - (\theta_{t+1} - \theta^*)^2] - \frac{\beta_{1,t}}{1-\beta_{1,t}} m_{t-1} \odot (\theta_t - \theta^*)
$$
$$
+ \frac{\alpha_t(1-\beta_1^t)\hat{m}_t^2}{2\sqrt{\hat{v}_t}(1-\beta_{1,t})}
$$

(32)

Similar to (Kingma & Ba, 2015), we use Young's inequality, $ab \leq a^2/2 + b^2/2$ and rearrange Eq. 32

$$
\nabla f_t(\theta_t) \odot (\theta_t - \theta^*) \leq \frac{(1-\beta_1^t)\sqrt{\hat{v}_t}}{2\alpha_t(1-\beta_1)}[(\theta_t - \theta^*)^2 - (\theta_{t+1} - \theta^*)^2] + \frac{\beta_{1,t}\sqrt{\hat{v}_{t-1}}}{2\alpha_{t-1}(1-\beta_{1,t})}(\theta^* - \theta_t)^2
$$
$$
+ \frac{\alpha_{t-1}\beta_1}{2(1-\beta_1)\sqrt{\hat{v}_{t-1}}} m_{t-1}^2 + \frac{\alpha_t(1-\beta_1^t)\hat{m}_t^2}{2\sqrt{\hat{v}_t}(1-\beta_1)}
$$
$$
\leq \frac{\sqrt{\hat{v}_t}}{2\alpha_t(1-\beta_1)}[(\theta_t - \theta^*)^2 - (\theta_{t+1} - \theta^*)^2] + \frac{\beta_{1,t}\sqrt{\hat{v}_{t-1}}}{2\alpha_{t-1}(1-\beta_{1,t})}(\theta^* - \theta_t)^2
$$
$$
+ \frac{\alpha_{t-1}\beta_1}{2(1-\beta_1)\sqrt{\hat{v}_{t-1}}} \hat{m}_{t-1}^2 + \frac{\alpha_t \hat{m}_t^2}{2(1-\beta_1)\sqrt{\hat{v}_t}}
$$

(33)

The regret is the sum of all the elements in $\nabla f_t(\theta_t) \odot (\theta_t - \theta^*)$ over all the iteration, where we have,

$$R(T) \leq \sum_{i=1}^{d} \frac{\sqrt{\hat{v}_{1,i}}}{2\alpha_1(1-\beta_1)}(\theta_{1,i} - \theta_{,i}^*)^2 + \sum_{i=1}^{d}\sum_{t=2}^{T} \frac{1}{2(1-\beta_1)}(\theta_{t,i} - \theta_{,i}^*)^2 \left( \frac{\sqrt{\hat{v}_{t,i}}}{\alpha_t} - \frac{\sqrt{\hat{v}_{t-1,i}}}{\alpha_{t-1}} \right)$$
$$+ \sum_{i=1}^{d}\sum_{t=1}^{T-1} \frac{\alpha_{t-1}\beta_1}{2(1-\beta_1)\sqrt{\hat{v}_{t,i}}}\hat{m}_{t,i}^2 + \sum_{i=1}^{d}\sum_{t=1}^{T} \frac{\alpha_t \hat{m}_{t,i}^2}{2(1-\beta_1)\sqrt{\hat{v}_{t,i}}} + \sum_{i=1}^{d}\sum_{t=1}^{T} \frac{\beta_{1,t}\sqrt{\hat{v}_{t-1}}}{2\alpha_{t-1}(1-\beta_{1,t})}(\theta_{,i}^* - \theta_{t,i})^2. \tag{34}$$

Apply Lemma A.3, we have

$$R(T) \leq \sum_{i=1}^{d} \frac{\sqrt{\hat{v}_{1,i}}}{2\alpha_1(1-\beta_1)}(\theta_{1,i} - \theta_{,i}^*)^2 + \sum_{i=1}^{d}\sum_{t=2}^{T} \frac{1}{2(1-\beta_1)}(\theta_{t,i} - \theta_{,i}^*)^2 \left( \frac{\sqrt{\hat{v}_{t,i}}}{\alpha_t} - \frac{\sqrt{\hat{v}_{t-1,i}}}{\alpha_{t-1}} \right)$$
$$+ \sum_{i=1}^{d} \frac{\alpha(1-\beta_2^t)(1-\beta_1^2)}{(1-\beta_2)^2(1-\beta_1^t)^2}G_\infty\sqrt{T} + \sum_{i=1}^{d}\sum_{t=1}^{T} \frac{\beta_{1,t}\sqrt{\hat{v}_{t-1}}}{2\alpha_{t-1}(1-\beta_{1,t})}(\theta_{,i}^* - \theta_{t,i})^2. \tag{35}$$

Note that, as pointed out by AMSgrad (Reddi et al., 2019), it requires the second moment estimate to increase monotonically, and they provide a modification to ensure that. From this perspective, our proposed method is similar and could also be modified for better convergence. From the assumption that $\|\theta_n - \theta_m\|_2 \leq D$, $\|\theta_n - \theta_m\|_\infty \leq D_\infty$ for any $n, m \in \{1, 2, \cdots, T\}$, we have

$$R(T) \leq \frac{D^2}{2\alpha(1-\beta_1)} \sum_{i=1}^{d} \sqrt{T\hat{v}_{T,i}} + \sum_{i=1}^{d} \frac{\alpha(1-\beta_2^t)(1-\beta_1^2)}{(1-\beta_2)^2(1-\beta_1^t)^2}G_\infty\sqrt{T} + \frac{D_\infty^2}{2\alpha} \sum_{i=1}^{d}\sum_{t=1}^{T} \frac{\beta_{1,t}\sqrt{t\hat{v}_{t-1,i}}}{(1-\beta_{1,t})}. \tag{36}$$

Since the gradient is bounded, for IO-Adam, we have $\sqrt{\hat{v}_{t,i}} < 2G_\infty$, therefore, we have

$$R(T) \leq \frac{dD^2G_\infty}{\alpha(1-\beta_1)}\sqrt{T} + \frac{\alpha d(1-\beta_2^t)(1-\beta_1^2)}{(1-\beta_2)^2(1-\beta_1^t)^2}G_\infty\sqrt{T} + \frac{dD_\infty^2 G_\infty}{\alpha} \sum_{t=1}^{T} \frac{\beta_{1,t}\sqrt{t}}{(1-\beta_{1,t})}. \tag{37}$$

For the last term, similar to the proof in (Kingma & Ba, 2015), we use arithmetic geometric series upper bound $\sum_t t\lambda^t \leq \frac{1}{(1-\lambda)^2}$ for the last term:

$$\sum_{t=1}^{T} \frac{\beta_{1,t}\sqrt{t}}{(1-\beta_{1,t})} \leq \sum_{t=1}^{T} \frac{1}{(1-\beta_1)}\lambda^{t-1}\sqrt{t}$$
$$\leq \sum_{t=1}^{T} \frac{1}{(1-\beta_1)}\lambda^{t-1}t$$
$$\leq \frac{1}{(1-\beta_1)(1-\lambda)^2} \tag{38}$$

Therefore, we have the following regret bound,

$$R(T) \leq \frac{dD^2}{\alpha(1-\beta_1)}G_\infty\sqrt{T} + \frac{\alpha d(1-\beta_2^t)(1-\beta_1^2)}{(1-\beta_2)^2(1-\beta_1^t)^2}G_\infty\sqrt{T} + \frac{dD_\infty^2 G_\infty}{\alpha(1-\beta_1)(1-\lambda)^2}. \tag{39}$$

$\square$

Based on Theorem A.4, it is easy to see that $R(T) = O(\sqrt{T})$, such that $\lim_{T\to\infty} \frac{R(T)}{T} = 0$, providing the same regret bound for our method IO-Adam as the Adam.

## B. Hyperparameter Details

### B.1. Experiments on GLUE

We follow the default setting in (Zhao et al., 2024) to conduct experiments. The learning rate is set at $3e - 5$ with batch size 16 and a linear learning rate scheduler. The model is fine-tuned on each task for 30 epochs. The rank for Galore and the buffer size for our method are set to 16.

### B.2. Experiments on MiniPile and C4

We use the code in (Zhao et al., 2024) to pretrain LLaMA-60M and LLaMA-130M on MiniPile. The learning rate is grid searched in the set $\{0.025, 0.02, 0.01, 0.002, 0.001\}$ for each optimizer while the batch size is set to 300. $\beta_1$ is set to 0.9 and $\beta_2$ is set to 0.999 as default for Adam, Galore and our method. Weight decay is set to 0.1. We set the number of warm-up steps to 600 for LLaMA-60M and 1000 for LLaMA-130M.

### B.3. Experiments on CIFAR10

We use the code in vit-pytorch to conduct our experiment. We set the patch size to 4, dimension to 128, depth to 6, number of heads to 16, and MLP dimension to 2048. All hyperparameters except the learning rate are set to the same. For learning rate, we use $5 \times 10^{-4}$ for AdamW, default $10^{-2}$ for Adafactor, and $10^{-3}$ for ours to let all optimizers perform well. The learning rate will be multiplied by 0.3 if the validation result becomes poor for 2 consecutive epochs. We run 50 epochs in total.

### B.4. Experiments on WikiText-2

We use the official example code from the Transformers library (Wolf et al., 2020) to conduct our experiments. We use default hyperparameters with a learning rate set at $5e - 4$. Since GPT-2 uses 1-dimensional convolution equivalent to a linear layer, we convert the code to use linear modules.

