# OpenReview forum: "IO-Adam: Rethinking Memory-Efficient Adaptive Optimizers from Gradient Computation"
_ICML.cc/2026/Conference — ICML 2026 regular_

### Official Review · Reviewer_BhmD · 2026-02-25

**Soundness:** 4
**Presentation:** 4
**Significance:** 4
**Originality:** 4
**Overall Recommendation:** 5
**Confidence:** 5

**Summary:**

This paper proposes an approximate formulation of second-order information in the Adam optimizer.
This approximation can reduce the storage, but increases the running time.
The authors show that their approach is the upper bound of Adam's second-order information,
  and derive an extended form by Holder's inequality to achieve a tighter bound by tuning the hyper-parameter $p$,
  which can improve the training performance.
The experiments present that their approach is competitive with AdamW in different scenarios.

**Compliance With Llm Reviewing Policy:**

Affirmed.

**Final Justification:**

I raise my decision to accept.

The rebuttal addressed my concern in the experiment part, which is the only weakness in my review.
Also, I re-think about the contributions of this paper.
This paper reduces the memory usage from 2409MB to 1599MB (around 34% reduction) in Table 2,
  but keeps an acceptable performance.
Therefore, I raise the score.

**Key Questions For Authors:**

Q1:
In the experiments, you only discuss
1. using different buffer sizes in Vision Transformer (Section 4.3)
2. using different $p$ in Section 4.4.
What if we consider them in all the experiments?

Q2:
The experiments shows that under the same setting, IO-Adam can have a smaller memory usage than AdamW.
What if we fix the memory size, but use different batch sizes or different model structures?

**Limitations:**

yes

**Strengths And Weaknesses:**

Strengths:

1. I do not check the convergence analysis in Section 3.4.2.
  However, other derivations are clear and easy to understand.
2. The paper utilizes a simple linear model to make the derivations easy to follow.
3. Memory issues in training are important when the training set is very large-scale.
  A larger model structure to improve the performance or a larger batch size to reduce the update steps in an epoch are the benefits after reducing the cost of memory.
4. I roughly check other approximations for reducing memory in using Adam.
  This work's idea is novel in itself.

Weakness:
There are some minor issues.
1. In eq. (8) and (10), modifying the font size is not a good thing.
2. In Table 1, it is hard to understand the meaning of $m$ and $n$ before reading Section 3.

---

> ### Author Rebuttal · Authors · 2026-03-31
>
> We sincerely thank the reviewer for the highly positive evaluation and the meticulous feedback on our manuscript's presentation and experimental depth.
>
> **Q1: Generalization of Buffer Size and Hyperparameter $p$**
>
> 1. In most of our experiments, the buffer size $r$ was set to align with the rank settings of GaLore to ensure a head-to-head, fair comparison of memory-efficiency and performance. As demonstrated in Section 4.3, **IO-Adam is notably insensitive to the specific choice of buffer size within a reasonable range**, maintaining stable performance across different configurations. We will include a more comprehensive ablation study across various architectures in the Appendix of the final version to further substantiate this robustness.
>
> 2. The parameter $p$ is a unique feature of IO-Adam derived from the Hölder's inequality framework. While it offers an additional dimension for theoretical adjustment, in practical applications, the default setting of $p=2$ is sufficient for most scenarios. We view $p$ as an optional enhancement rather than a necessary tuning requirement. To provide a clearer picture, we will add more experimental results involving different $p$ values in the revised manuscript.
>
> **Q2: Fixed memory budget with larger batch sizes or models.**
>
> Modern large-scale training typically utilizes Gradient Accumulation (multiple forward/backward passes per update). By significantly reducing the optimizer's memory footprint, IO-Adam allows for either (a) larger per-device batch sizes or (b) more accumulation steps within the same VRAM limit, both of which can lead to higher training stability and throughput.
>
> We believe the optimization quality remains consistent while the hardware efficiency improves. Since the actual speed-up and memory ceiling are highly hardware-dependent, IO-Adam provides a lower "memory floor" that benefits diverse GPU configurations. We will provide more specific profiles comparing throughput and max-batch-size under fixed memory budgets in the final version.
>
> **Minor Issues (Fonts and Definitions)**
>
> We appreciated your advice for the inconsistent font sizes in Eq. (8) and (10) and the lack of clarity in Table 1. We will:Standardize all formula font sizes.
>
> Add explicit definitions for $W_{in}$ and $W_{out}$ in the caption of Table 1 or the preceding text to ensure the table is self-explanatory.
>
> We appreciate the reviewer's insightful comments. We look forward to further discussions and are happy to provide any additional clarifications if needed.

---

> > ### Author Rebuttal · Reviewer_BhmD · 2026-04-02
> >
> > The authors claim that they will add some experiments in the paper.
> > However, because I do not see the results in the reply back, it is hard to have more information to review this paper.
> > Thus, I keep my original decision.

---

> > > ### Author Response · Authors · 2026-04-05
> > >
> > > Dear Reviewer BhmD,
> > >
> > > We sincerely thank the reviewer for the follow-up feedback. We understand the importance of providing empirical evidence to validate our theoretical claims. To address concerns about the sensitivity of the buffer size (l) and the power parameter (p), we have conducted additional experiments with ViT models on the CIFAR-10 dataset. The results are summarized in the table below. All training configurations remain identical to those described in the paper, except that these experiments were executed on an RTX 4090 GPU. (AdamW optimizer leads to a `73.8%` test accuracy.)
> > >
> > > Table: Accuracy (%) under different combinations of buffer size (`l`) and power parameter (`p`)
> > >
> > > | l/p | p=1.4 | p=1.6 | p=1.8 | p=2.0 | p=2.2 | p=2.4 | p=2.6 |
> > > | --- | ----- | ----- | ----- | ----- | ----- | ----- | ----- |
> > > | l=1 | 70.61 | 72.61 | 74.03 | 74.85 | 75.84 | 76.05 | 76.27 |
> > > | l=2 | 70.61 | 72.60 | 74.02 | 74.86 | 75.85 | 76.06 | 76.25 |
> > > | l=8 | 70.60 | 72.55 | 74.00 | 74.86 | 75.86 | 76.01 | 76.24 |
> > > | l=4 | 70.63 | 72.59 | 74.03 | 74.85 | 75.86 | 76.02 | 76.23 |
> > >
> > >
> > > As shown in the table, for any fixed value of `p`, the performance remains remarkably stable across different buffer sizes (from l=1 to l=8). This confirms our claim in Section 4.3 that IO-Adam is insensitive to the buffer size.
> > >
> > > Regarding the hyperparameter "p", different tasks may require different optimal values of "p". However, we want to emphasize that tuning "p" is not a mandatory prerequisite. Almost all of our experiments are conducted with $p=2$, showing that IO-Adam does not necessarily require tuning "p" for better performance.
> > >
> > > We remain committed to including a unified and comprehensive ablation study in the final version of the paper and hope these additional results provide the necessary information to resolve your concerns. We sincerely appreciate your thoughtful comments and efforts. We look forward to hearing from you regarding whether your concerns have been fully addressed.
> > >
> > > Best regards,
> > >
> > > Authors

---

### Official Review · Reviewer_q653 · 2026-03-04

**Soundness:** 3
**Presentation:** 3
**Significance:** 2
**Originality:** 3
**Overall Recommendation:** 4
**Confidence:** 3

**Summary:**

In this paper, the authors propose a new approach for a memory efficient adaptive learning rate by exploiting the fact that gradients of a matrix is a Kronecker product of inputs and gradients of outputs. Experiments with BERT LM, LLM, and ViT shows that the proposed approach can reduce memory by up to 30% and yield on par performance as AdamW.

**Compliance With Llm Reviewing Policy:**

Affirmed.

**Key Questions For Authors:**

1. Please consider including the comparison to Shampoo and Soup.
2. Please consider including the experimental results for a larger size LLM.

**Limitations:**

Please consider including a discussion regarding how the proposed method may lose important structural information in gradients, which is picked up by second methods such as Shampoo and Soup.

**Strengths And Weaknesses:**

Strength:
1. The idea of leveraging the structure of matrix gradients for memory efficient adaptive learning rate is novel.
2. Empirical studies with BERT LM, LLM, and ViT showed that the proposed method is able to yield similar performance as AdamW with significantly reduced memory.

Weakness:
1. The authors should compared to other more effective approaches for approximating the Hessian matrix, such as Shampoo and Soup, which are in general more effective than AdamW.
2. The proposed method essentially lose the structure of the original gradient (e.g. the approximation is essentially of rank one matrix). Although only the individual elements in the approximation scheme are used for adaptive learning rate, losing the structural information of gradients is still concerning to me (e.g. rank one matrix will make elements in the matrix to be strongly correlated, which may affect the impact of adaptive learning rates).
3. It would be nice to examine the effect of the proposed approach for a larger size model, particularly for LLM.

---

> ### Author Rebuttal · Authors · 2026-03-31
>
> We thank the reviewer for the highly professional and insightful comments, particularly regarding the structural properties of our approximation and the scalability of IO-Adam.
>
> **Q1: Comparison with Shampoo and SOAP.**
>
> We appreciate the opportunity to clarify the relationship between IO-Adam and methods like Shampoo and SOAP. Shampoo-type methods focus on preconditioning to accelerate convergence using Kronecker-factored approximations of the Hessian. In contrast, IO-Adam is designed for extreme memory efficiency. Our core motivation is to provide a memory-lean alternative to Adam by rethinking the second-moment estimation.
>
> We hope the gradient decomposition perspective provided in our paper can also innovate future works on optimizers like shampoo and SOAP.
>
> **Q2: Low-rank structure**
>
> The reviewer's concern regarding the rank of the second-moment estimation is well-taken. We clarify our logic as follows:
>
> 1. Our method builds on the core observation that the weight gradient is the product of the input and the output gradient. A parameter would generally recieve a large gradient if it generally recieves large input and large gradient on its corresponding output. Therefore，we adjust the learning rate by tracking the magnitudes of input ($X$) and output gradients ($\delta$）.
> 2. While the basic formulation implies a rank-one structure, in practice, we introduce a buffer mechanism (similar to GaLore) that allows the rank of the approximation to be adjusted. This ensures that we are not strictly limited to a rank-one approximation, allowing the model to capture more structural information when memory permits.
>
>
> **Q3: Experiments on larger LLMs.**
>
> We appreciate the suggestion to include more large-scale LLM results.
>
> The hyperparameters and model scales in our current submission follow the standard protocols established in previous memory-efficient optimizer literature.
>
> We acknowledge the importance of broader validation. Preliminary results indicate that IO-Adam maintains its memory advantage while achieving performance parity with Adam in LLM fine-tuning tasks. Due to the limited time, we will provide additional fine-tuning results to further demonstrate IO-Adam’s scalability promptly during discussion period. These comprehensive profiles will be included in the final version.
>
>
> We appreciate the reviewer's insightful comments. We look forward to further discussions and are happy to provide any additional clarifications if needed.

---

### Official Review · Reviewer_FUKA · 2026-03-05

**Soundness:** 2
**Presentation:** 2
**Significance:** 2
**Originality:** 2
**Overall Recommendation:** 3
**Confidence:** 3

**Summary:**

This paper presents an adaptive moment estimation by input and output gradient (IO-Adam) for training deep neural networks. It provides numerical results indicating that IO-Adam performs better than Adam-type algorithms.

**Compliance With Llm Reviewing Policy:**

Affirmed.

**Final Justification:**

I checked the replies on my concerns. However, I think more theoretical results (e.g., convergence analysis, convergence rate analysis, and computational complexity, and so on)  are needed to support the numerical results. I still think $R(T)$ in (16) may need to be redefined. Or, I suggest the authors use the squared norm of the full gradient and the distance between the value of the objective function and the optimal value instead of $R(T)$. Therefore, I keep my score.

**Key Questions For Authors:**

1. Proposition 3.1 indicates that IO-Adam has the same convergence property as AMSgrad. I think that only the proposition is insufficient to support the numerical results. For example, can we prove that IO-Adam converges faster than Adam-type algorithms?
2. Proposition 3.1 uses the performance measure (16). I have some concerns about $R(T)$ defined by (16). For example, let us consider the empirical risk minimization, i.e., the case where we know the number of training samples, denoted by $n$. In the case, we have $n$ loss functions, $f_1, \cdots, f_n$. Then, we can define $R(n) = \sum_{t=1}^n [f_t (\theta_t) - f_t (\theta^*)]$, however, we cannot define $R(n+1), R(n+2), \cdots$ (since we cannot define $f_{n+1}, f_{n+2}, \cdots$). I think $R(T)$ in (16) may need to be redefined.

**Strengths And Weaknesses:**

**Strengths:**
- This paper explains the first and second moment estimations (Sections 3.2 and 3.3) and the connection and difference between IO-Adam and Adam (Section 3.4).
- This paper presents numerical results indicating that IO-Adam performs better than Adam-type algorithms (Section 4).

**Weaknesses:**
- I think the numerical results have contributions. Meanwhile, I am not satisfied with theoretical results (Proposition 3.1).
- I think that more theoretical results to support the numerical results are needed.

---

> ### Author Rebuttal · Authors · 2026-03-31
>
> We thank the reviewer for the rigorous evaluation of our theoretical results. We provide the following clarifications regarding Proposition 3.1 and the notation in Eq. (16).
>
> **Q1: Convergence speed vs. Adam-type algorithms.**
>
> We appreciate the reviewer's observation. We would like to clarify that the primary objective of IO-Adam is **memory efficiency rather than establishing a theoretically superior convergence rate** over Adam-type algorithms. Our theoretical analysis aims to demostrate that IO-Adam preserves the $O(1/\sqrt{T})$ convergence rate consistent with Adam/AMSGrad while significantly reducing the memory footprint for second-moment estimation. We consider the formal derivation of a theoratically faster convergence rate as a compelling direction for future work.
>
> **Q2: Clarification on the definition of $f_t$ and $g_{t,i}$ in Eq. (16).**
>
> The notation in Eq. (16) follows the standard Online Convex Optimization (OCO) framework, which is the conventional setting for analyzing optimizers like Adam and AMSGrad. We acknowledge the reviewer's point regarding the behavior under specific finite constraints. Following the established literature in this domain, our current analysis focuses on the asymptotic properties where data approaches infinity.
>
> Could you clarify the points for unsatisfiabiliity to our theoretical results? We look forward to addressing this in future extensions of this work.
>
> We appreciate the reviewer's insightful comments. We look forward to further discussions and are happy to provide any additional clarifications if needed.

---

> > ### Author Rebuttal · Reviewer_FUKA · 2026-04-05
> >
> > Thank you for your replies. However, I have some concerns.
> > - I think that theoretical results are needed to support the numerical results. The authors said that IO-Adam significantly reduces the memory footprint for second-moment estimation. So, I would like to suggest that the authors provide a theoretical guarantee showing that IO-Adam reduces memory usage more effectively than Adam-type algorithms.
> > - I am sorry that I do not fully understand your comments. I still think (16) may need to be redefined.

---

> > > ### Author Response · Authors · 2026-04-05
> > >
> > > Dear Reviewer FUKA,
> > >
> > > Thanks for the further feedback. In the following, we address each of your further comments.
> > >
> > > >  I would like to suggest that the authors provide a theoretical guarantee showing that IO-Adam reduces memory usage more effectively than Adam-type algorithms.
> > >
> > > As discussed in Table 1 and demonstrated in Figure 1, our method reduces memory usage by separately tracking the input and output for second-moment estimation.
> > >
> > > Consider a weight matrix $W \in \mathbb{R}^{n \times n}$, Adam-type algorithms store first and second moment estimation of the same size as the weight matrix, resulting in a space complexity of $O(n^2)$ for the optimizer state. In this paper, we show that the weight gradient is actually the product of the input $x\in \mathbb{R}^{n}$ and the output gradient $\nabla_{y} \mathcal{L} \in \mathbb{R}^n$. Therefore, we propose replacing the second-moment estimate of the weight gradient (a matrix) with the product of the second-moment estimates of the input and output gradients (two vectors). It results in $O(n)$ in space complexity.
> > >
> > > > I am sorry that I do not fully understand your comments. I still think (16) may need to be redefined.
> > >
> > > Eq. 16 follows previous work [1,2], in which the framework aims to prove a regret bound in an online learning setting. We want to emphasize that the topic of this paper is not the development of a new framework for analyzing the regret bounds of optimizers.
> > >
> > > [1] Zinkevich, Martin. Online convex programming and generalized infinitesimal gradient ascent. 2003.
> > >
> > > [2] Diederik P. Kingma, Jimmy Ba. Adam: A Method for Stochastic Optimization. ICLR (Poster) 2015
> > >
> > > We hope our reply can address your concerns, and we look forward to your reply.
> > >
> > > Best,
> > >
> > > Authors

---

### Official Review · Reviewer_FWMh · 2026-03-10

**Soundness:** 2
**Presentation:** 3
**Significance:** 2
**Originality:** 3
**Overall Recommendation:** 4
**Confidence:** 3

**Summary:**

The paper proposes to improve the memory efficiency in Adam via more efficient computation of v_t. The idea is to save the squared representations at each layer and multiply them by the squared output errors to get the estimated second moment, leading to significantly smaller memory requirements. The authors show that Adam-IO has similar convergence guarantees as Adam. Numerical simulations are present as well.

**Compliance With Llm Reviewing Policy:**

Affirmed.

**Final Justification:**

I thank the authors for the rebuttal. However, in the current form, I believe the paper would still benefit from a more systematic investigation of the memory gains and associated runtime costs. In addition, the runtime increase compared to vanilla AdamW is relatively large, as reported in the last author's response. Consequently, I believe that "weak accept" is the fair score. Therefore, I will keep it.

**Key Questions For Authors:**

See "strengths and weaknesses"

**Limitations:**

Yes

**Strengths And Weaknesses:**

Presentation: The paper is well written and easy to follow

Significance: The work tackles an important problem and attempts to find a way to reduce the memory usage of Adam. In addition to that, the Hölder-inequality-based modification is interesting. At the same time, it is difficult to assess the practical utility, since the largest even for the largest LLM analyzed in the paper, GPT-2, the memory consumption is not reported.

Soundness:
Currently, the memory gains are only visible in some of the experiments, while in others, as mentioned above,  (LLaMA on C4 and MiniPile, and GPT-2 on WikiText) remain unreported. In addition, the total runtime, as shown in one experiment, is higher compared to other baselines. Lastly, the model is applicable to only linear layers.

Could the authors report the memory consumption and the runtime in the missing experiments? Furthermore, how this approach can be extended to other types of layers (2d convolution, attention).

Lastly, the authors mention that Adam-IO generally requires a higher lr due to overestimated v_t. A practical guidance on how to choose the lr in different settings would be helpful. In the "Pretraining LLaMA on C4 and MiniPile" experiment, it is mentioned that the lr of Adam-IO should be an order of magnitude higher than that of Adam. Is it always the case? If no, can the author add a paragraph describing how to choose the lr?

Originality: the work introduces an original idea of how to obtain the second moment estimate more efficiently. Although it is only applicable to linear layers, and more thorough empirical validation is necessary, I find this idea novel and promising.

---

> ### Author Rebuttal · Authors · 2026-03-31
>
> We thank the reviewer for the insightful comments.
>
> **Q1: Memory consumption/Runtime on all the experiments**
>
> Thank you for your valuable advise. Currently, we only provide partial results due to the limited space to demonstrate the efficiency of our method.
>
> Our method multiply two small matrices for the second moment estimation where one matrix record the input information and the other record the output gradient information, which is similar to LoRA. The memory consumption of our method is similar to previous works on memory efficient optimizers like [Galore](https://arxiv.org/abs/2403.03507). Regarding the runtime, our method involves recording the input and output gradient. While they are available during back propagation, we have to use hooks to record them when using pytorch, which introduces a small latency as shown in Table 6.
>
> We will include memory consumption and total runtime for each experiment in the final version to provide a complete profile of IO-Adam's efficiency.
>
> **Q2: Extension to other type of layers**
>
> The fundamental principle of IO-Adam is to optimize memory from the perspective of gradient computation. This logic is universally applicable to any operator that can be formulated as or transformed into a matrix multiplication. It include generally all the regular linear operators.
>
> Since modern deep learning libraries (e.g., cuDNN) implement Conv2d using the im2col transformation to leverage GEMM, the gradient computation for convolutional weights follows the same algebraic structure as Linear layers. Thus, our memory-saving strategy for the first and second moments directly extends to Conv2d by rewriting the operator in its equivalent matrix form.
>
> **Q3: Explanation of Learning Rate (LR) selection.**
>
> For fair comparison, we apply learning rate grid search for all the optimizers. As detailed in Appendix B of our manuscript, all learning rates were determined through a rigorous grid search to ensure a fair comparison.
>
> In practice, the optimal learning rate for IO-Adam is typically 10 times that of standard Adam. This empirical heuristic makes IO-Adam easy to deploy without exhaustive tuning, as it scales predictably relative to the baseline.
>
> We appreciate the reviewer's insightful comments. We look forward to further discussions and are happy to provide any additional clarifications if needed.

---

> > ### Author Rebuttal · Reviewer_FWMh · 2026-04-03
> >
> > I thank the authors for addressing my comments. The Q3 is addressed. However, some of the concerns remain
> > 1. While the authors commit to reporting the total memory and runtime for each experiment, the actual results are missing so far
> > 2. The discussion on how to extend IO-Adam to attention layers is missing

---

> > > ### Author Response · Authors · 2026-04-05
> > >
> > > Dear Reviewer FWMh,
> > >
> > > We sincerely appreciate your further comments and the effort you devoted to the review process. In the following, we address each of your further concerns:
> > >
> > > > No actual results on runtime and memory usage provided so far.
> > >
> > > We are currently running our experiments to collect runtime results. Due to our current experimental conditions, the execution time of our deep learning experiments may fluctuate depending on the task load and the status of the computing cluster we are using. We need multiple runs for a fair comparison on runtime. In the following, we report the total runtime we record for training ViT-Base on CIFAR10 with H200.
> > > | Optimizer  | Runtime   |
> > > | ---------- | --------- |
> > > | Ours(l=16) | 42min 27s |
> > > | Ours(l=8)  | 53min 57s |
> > > | Ours(l=4)  | 57min     |
> > > | Ours(l=2)  | 52min 20s |
> > > | Ours(l=1)  | 46min 57s |
> > > | AdamW      | 34min 34s |
> > > | Adafactor  | 49min 51s |
> > >
> > > Here, l is the buffer size used in our proposed IO-Adam. While the runtime may fluctuate, our method generally incurs overhead comparable to that of existing memory-efficient optimizers.  We will include more results on the runtime and total memory usage of our proposed method.
> > >
> > > We would like to take this opportunity to further analyze the memory usage and runtime of our proposed method. In our proposed method, we use a low-rank approximation for second-moment estimation in Adam, with one matrix storing the input and the other storing the output gradient. Regarding memory usage, it introduces memory usage similar to that of existing memory-efficient optimizers. Since many existing optimizers also use low-rank approximation for second-moment estimation, the main difference between our method and existing memory-efficient optimizers lies not in memory usage but in how we approximate the second moment, which we approach from a new perspective based on gradient computation.
> > >
> > > Regarding the total runtime of our proposed method, it introduces overhead similar to that of previous optimizers like Adafactor, which collect additional information from the gradient. However, the input and output gradients we store are already computed via backpropagation. As discussed in the paper, the main time cost comes from saving buffers and handling hooks for the input and output gradients of each linear layer.  Therefore, we seek to implement our algorithm at a lower, hardware-proximal level to improve execution efficiency in the future.
> > >
> > > > The discussion on how to extend IO-Adam to attention layers is missing
> > >
> > > Thank you for bringing up the good question. While attention layers are widely adopted in many architectures (especially in transformers), they consist of linear layers projecting the inputs into queries, keys and values. Therefore, one can apply our method to attention layers by applying our method to each of the linear layers consisting the attention module. It is exactly what we do in our experiments where we apply our method to layers like "q_proj", "k_proj", 'v_proj" in the attention modules  of transformers.
> > >
> > > We hope our reply could address your concerns. We really look forward to your further comments. Thanks again for your valuable feedback.
> > >
> > > Best,
> > >
> > > Authors

---

### Decision · Program_Chairs · 2026-04-30

**Decision:**

Accept (regular)

**Comment:**

The paper focuses on designing memory-efficient optimizers. Unlike previous works that use low-rank approximations to save memory, the authors propose tracking input and output gradients to efficiently estimate moments. They also demonstrate that this approach can save around 30% of memory. I find the idea interesting and therefore recommend acceptance.